# StructPolicy: Robust Imitation Learning Policy Guided by Structure Map

## Abstract

Imitation Learning (IL) offers an effective approach for robot manipulation by learning a mapping function from visual inputs to actions. However, this paradigm suffers from domain discrepancies and is highly sensitive to distribution shifts, as the mapping function inherent in IL tends to overfit task-irrelevant visual noise, thereby compromising robustness. To address this, we propose StructPolicy, a lightweight module that guides the robot to acquire structural knowledge by constructing a stable and task-relevant Structure Map. The structure map is composed of task-relevant object structures and reveals the topological cues essential for accomplishing manipulation tasks. By filtering out visually distracting noise and retaining only the structural attributes, it effectively guides the robot's policy learning toward robust manipulation. We begin by introducing a general object structure representation based on atomic geometric primitives, enabling flexible composition and scalability to arbitrary objects. Building on this, we design StructGen, a module that automatically constructs structure maps from visual observations. Finally, we design StructTransformer, which employs hierarchical attention over structure vertices to extract map features from the structure map for action prediction. We extensively evaluate StructPolicy across 3 IL models, 3 different simulators, and 50 diverse manipulation tasks. In addition, through ablation studies under various visual changes, including visual noise, camera viewpoint shifts, light intensity, and background color variations, we demonstrate that StructPolicy consistently improves robustness against distribution shifts. Results demonstrate consistent and significant performance improvements across all tasks, validating the effectiveness and robustness of StructPolicy. Our code will be released publicly to facilitate reproducibility and further research.

## 1 Introduction

Imitation Learning (IL) has emerged as a widely adopted paradigm in robot manipulation due to its efficiency, ease of training, and strong performance in many tasks (Schaal et al., 2003; Ross et al., 2010; Lin et al., 2023; Kim et al., 2021; Zhao et al., 2023). By learning directly from expert demonstrations, IL bypasses the need for handcrafted rewards and allows agents to acquire complex manipulation skills in a supervised manner. These benefits have made IL attractive for both simulation and real-world applications.

However, IL typically learns a mapping function from observations to actions, without modeling the underlying physical structure or object-level semantics of the environment. This lack of structured understanding makes IL models fragile when deployed in environments that deviate from the training distribution. One key failure mode is known as **domain discrepancy** (Zare et al., 2024), referring to the mismatch between training and deployment settings. For example, in a push-block task, a policy trained on one tabletop setup may struggle when tested in another with slightly different object positions or textures. The robot may fail to locate the target block correctly or produce actions misaligned with the new context, as it relies heavily on pixel-level visual cues rather than topological relationships. This significantly limits the generality and robustness of IL in real-world robot manipulation.

Previous studies have attempted to address this issue from two main perspectives. One line of work focuses on enhancing the observation representation by leveraging RGB-D images, multi-view

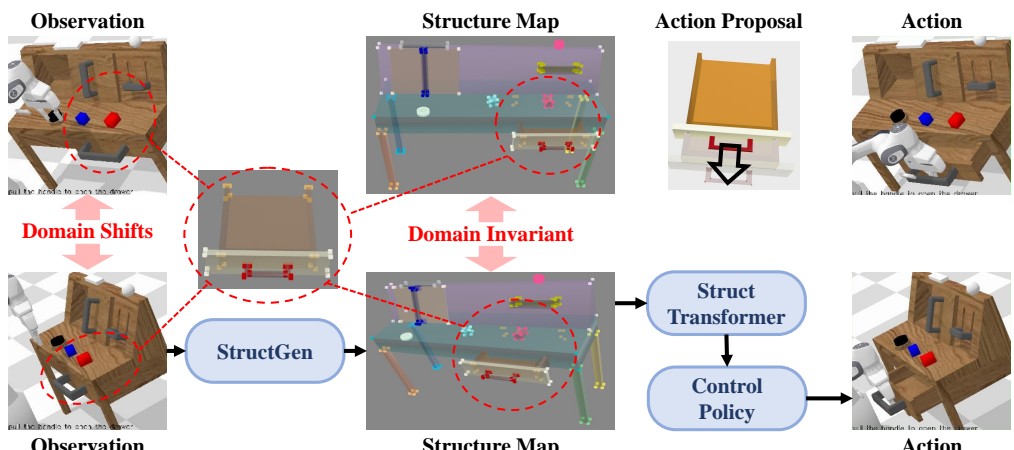

Figure 1: StructPolicy mitigates domain shifts between training and testing by constructing domain-invariant structure maps. The structure map encodes object-level structures that remain consistent across different camera viewpoints, thereby enhancing robustness to visual variations.

cameras, or point clouds, aiming to improve policy robustness by injecting more geometric information (Jangir et al., 2022; Chen et al., 2021; Akinola et al., 2020; Yan et al., 2018). The other line introduces an implicit feature space learned by visual encoders, which attempts to suppress visual noise through end-to-end training (Simeonov et al., 2022; Ryu et al., 2024). While these approaches show some improvements, the former remains confined to the visual level and fails to fundamentally resolve the issue of domain discrepancies. The latter suffers from poor interpretability and lacks explicit structural reasoning, making it difficult for the robot to truly learn the key factors critical to manipulation.

In contrast, we propose **StructPolicy**, which introduces a domain-invariant **Structure Map** that explicitly encodes both the intrinsic structures of task-relevant objects and the topological relationships among them. As illustrated in Figure 1, this structured representation filters out task-irrelevant visual noise and preserves only manipulation-critical knowledge, thereby guiding the policy to focus on stable and meaningful cues. By grounding policy learning in structural and topological information, StructPolicy significantly enhances the robustness of imitation learning against visual and environmental variations.

To construct the structure map across diverse manipulation scenarios, we decompose objects into a small set of atomic geometric primitives. These canonical primitives are parameterized and can be flexibly composed and scaled, enabling the representation of arbitrary objects and their interactions.

StructPolicy consists of two main modules: StructGen and StructTransformer. StructGen is a module that constructs structure maps from task-conditioned observations. Given an RGB image and a task description, StructGen employs Grounded-SAM to localize task-relevant objects, and then queries the ConceptFactory (Sun et al., 2025) database to retrieve corresponding parameterized templates. In parallel, the observation is processed by a visual encoder to extract features, which are fed into a parameter estimator to predict the geometric parameters of the retrieved templates. These parameters are then used to instantiate the templates with atomic geometric primitives, yielding the complete structure map. This pipeline enables StructGen to produce structure maps in an end-to-end manner, encoding both geometric and topological regularities with scalability and adaptability to diverse manipulation scenarios.

StructTransformer is a transformer-based encoder that extracts map features from the structure map. Given the structure map, the pipeline proceeds in two parallel streams. The target pose is encoded through a lightweight target pose encoder, producing target pose features, while the structure map is first processed at the primitive level, where each primitive is passed through a StructTransformer Block to extract its primitive token. These tokens are then further aggregated by another Struct-Transformer Block to capture inter-primitive topological relationships, resulting in the final global features. These two streams are concatenated into a unified map feature, which serves as the final

input to the control policy. This design enables the model to jointly reason about object topology and task-specific pose information, thereby highlighting manipulation-critical cues.

Importantly, StructPolicy is designed as a lightweight plug-in module that can be seamlessly integrated into existing IL models without modifying the backbone architecture. It introduces few inference overheads while providing explicit structural guidance.

Our contributions can be summarized as follows,

- We propose **StructPolicy**, a lightweight yet effective module for imitation learning that enhances robustness by guiding the robot to construct a consistent and task-relevant **structure map**, thereby mitigating domain discrepancies and distribution shifts.
- We develop **StructGen**, an automated structure map generation method that leverages atomic geometric primitives to flexibly compose and scale to arbitrary objects, enabling accurate representation of large and complex scenes.
- We design **StructTransformer**, a transformer encoder that employs global attention over structure vertices to extract structural features from the structure map for action prediction, achieving both high effectiveness and fast inference.
- We conduct comprehensive evaluations across 50 manipulation tasks in 3 simulation environments with 3 state-of-the-art IL models, showing consistent performance gains. Further ablation studies under 4 kinds of visual changes confirm that StructPolicy substantially improves robustness against distribution shifts while maintaining strong overall effectiveness.

## 2 RELATED WORKS

### 2.1 IMITATION LEARNING FOR ROBOT MANIPULATION

Imitation Learning (IL) provides a simple yet effective paradigm for robot manipulation by learning a direct mapping from observations to actions. Recent advances have introduced powerful generative models into this paradigm, such as Diffusion Policy (Chi et al., 2024) using diffusion models, FlowPolicy leveraging flow matching, and IMLE Policy (Rana et al., 2025) based on implicit maximum likelihood estimation, all of which have shown promising performance. Other approaches like Lift3D (Jia et al., 2025), PolarNet (Chen et al., 2023), and Perceiver-Actor (Shridhar et al., 2022) enhance IL by incorporating RGB-D information or improved spatial encoders to strengthen the model's 3D perception. However, these methods largely operate at the visual level, remaining fundamentally limited by the noise and variability inherent in complex visual scenes.

In contrast, our proposed StructPolicy tackles this issue by explicitly guiding the robot to learn a domain-invariant structure map. This structured representation encodes stable object-level geometry, enabling the policy to focus on task-relevant structural cues and significantly improving robustness in the face of visual distractions.

### 2.2 STRUCTURE-AWARE SHAPE REPRESENTATIONS

Structure representations play a crucial role across various domains in Computer Vision and Robotics, particularly in tasks that require 3D understanding. Unlike raw visual observations such as RGB images or point clouds, structure-based representations provide higher-level abstraction and greater stability, making them especially valuable for complex reasoning, long-term planning, and informed decision making. Structural cues—such as part layouts, skeletons, and spatial topologies—have been successfully applied in diverse areas including 3D shape generation (Mo et al., 2019), 3D reconstruction (Zhang et al., 2025), and keypoint detection (Shi et al., 2021).

A prominent line of work abstracts 3D structure through compositions of primitive shapes. For example, (Tulsiani et al., 2016) assembles objects using cuboids via convolutional networks, while (Zou et al., 2017) adopts probabilistic recursive neural networks (PRNN) for structural parsing. To go beyond cuboids, (Paschalidou et al., 2019) introduces superquadric primitives, and (Paschalidou et al., 2020) proposes a hierarchical decomposition into parts for better geometric fidelity.

In contrast, we construct object structure using a compact set of 10 atomic geometric primitives, which can be flexibly composed and scaled to represent a wide range of objects in manipulation

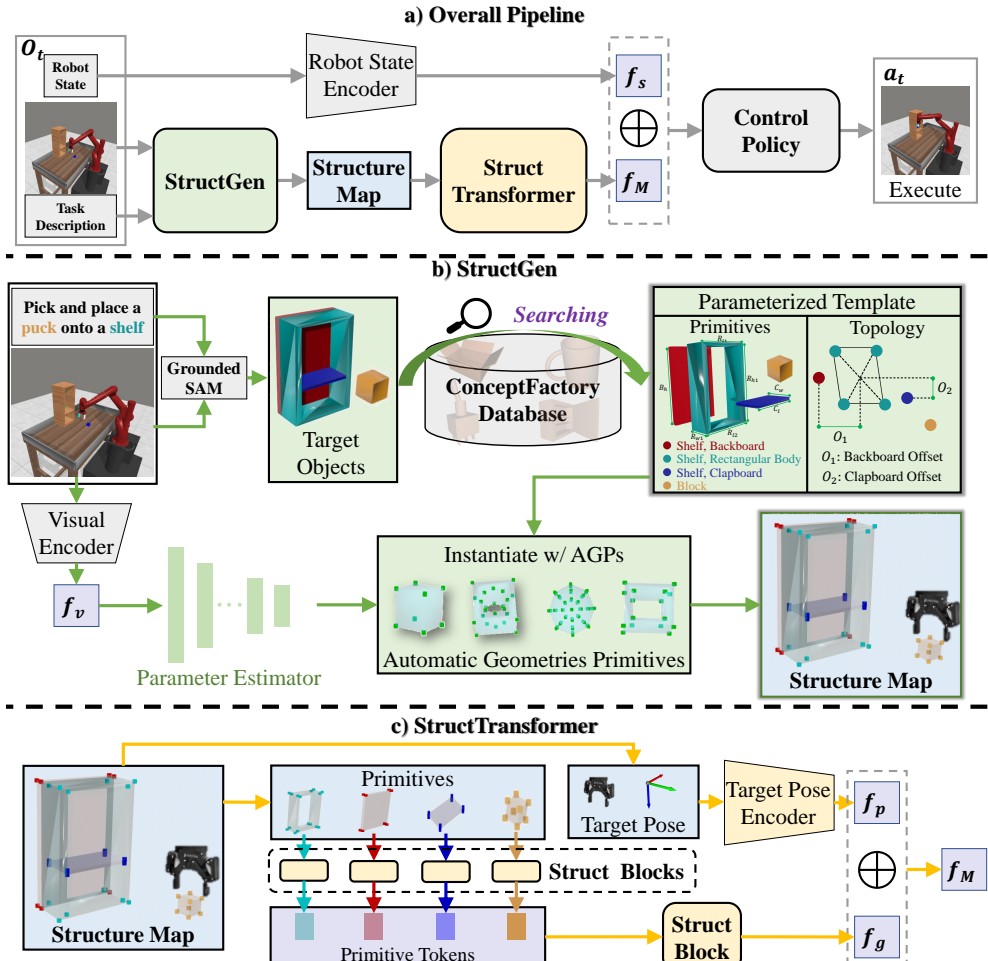

Figure 2: Overview of StructPolicy. *a)* StructPolicy first guides the robot to construct a structure map via StructGen, then uses StructTransformer to extract map features $f_M$ for action prediction. *b)* StructGen constructs structure maps from task-conditioned observations by localizing relevant objects and instantiating parameterized primitives. *c)* StructTransformer encodes the structure map with StrutTransformer Block(Struct Block) to extract global features $f_g$ and integrates them with the target pose features $f_p$, producing map features $f_M$.

scenarios. This formulation not only ensures sufficient expressiveness but also offers ease of parameterization and integration into downstream policy learning.

## 3 METHOD

### 3.1 OVERVIEW

Overview pipeline of StructPolicy is illustrated in Figure 2.

Most imitation learning pipelines follow a standardized structure, typically consisting of two components: a visual encoder that extracts meaningful features from observations and a control policy that maps these features, often combined with the robot state features, to predicted actions. StructPolicy augments the imitation learning pipeline with a structure-aware module that guides the robot to acquire robust structural knowledge from task-conditioned observations. It consists of two components: StructGen, which automatically constructs structure maps from RGB images and task descriptions using atomic geometric primitives to capture both object-level geometry and inter-object topology; and StructTransformer, which first derives primitive tokens and global features through

StructTransformer Blocks, and then concatenates the global features with target pose features to obtain the final map features.

At the core of StructPolicy is the structure map, a set of labeled vertices representing the 3D positions and structural attributes of task-relevant objects. This map captures both intra-object geometry and inter-object topology, providing a compact abstraction of the scene. To construct such maps, we design StructGen, an automated method that localizes task-relevant objects from RGB images and task descriptions, retrieves parameterized templates from ConceptFactory database, and instantiates them with atomic geometric primitives to capture geometric and topological regularities. On top of this, we develop StructTransformer, which derives primitive tokens and global features through StructTransformer Blocks and further integrates the global features with target pose features, yielding manipulation-critical map features.

Finally, the map features are concatenated with the robot states features to form the input to the control policy. This embedding allows the policy to benefit from structural knowledge, which provides object-level reasoning and stability. Importantly, StructPolicy functions as a plug-in module and requires no modification to the backbone policy architecture, ensuring compatibility with a wide range of existing IL models.

## 3.2 STRUCTURE MAP

We formulate the structure map in three levels: atomic geometric primitives, object structure, and the final structure map.

**Atomic Geometric Primitives**  We begin with a library of 10 atomic geometric primitives (AGPs): *Cuboid*, *Sphere*, *Cylinder*, *Triangular Prism*, *Cone*, *Rectangular Ring*, *Ring*, *Torus*, *Box-Cylinder-Ring*, and *Cylinder-Box-Ring*, adapted from (Sun et al., 2025). To construct structural representations, each AGP is discretized into labeled 3D points using two strategies. For polyhedral primitives (e.g., Cuboid, Triangular Prism), we apply triangular surface decomposition by uniformly triangulating the surface and taking triangle vertices as structure points. For curved primitives (e.g., Cylinder, Torus), we use uniform circular sampling, where points are sampled on planar arc sections with a resolution parameter $N$. These discretized point sets serve as reusable templates for building more complex structures. Detailed design is shown in Appendix D

**Object structure**  An object $o$ is represented as a composition of multiple AGPs. Formally, its object structure is defined as the union of the discretized point sets of its constituent primitives:

$$\mathcal{S}_o = \bigcup_{m=1}^{M_o} \{(\mathbf{p}_i, \ell_i)\}_{i=1}^{N_m},$$

where $\mathbf{p}_i \in \mathbb{R}^3$ is a sampled 3D point and $\ell_i$ indicates the type of atomic geometric primitive (AGP). This formulation enables diverse objects to be expressed by combining and arranging AGPs.

**Structure map**  Given a task environment with $K$ objects, the structure map is defined as the collection of their object structures:

$$\mathcal{M} = \{\mathcal{S}_{o_j}\}_{j=1}^K = \{(\mathbf{p}_n, \ell_n, o_n)\}_{n=1}^{N_M},$$

where $o_n$ indexes the object instance and $N_\mathcal{M} = \sum_{j=1}^K N_{o_j}$. Thus, the structure map encodes both intra-object geometry and inter-object topology in a unified, compositional representation.

## 3.3 STRUCTGEN

StructGen is responsible for automatically constructing structure maps from task-conditioned observations. Given an RGB image and a task description, StructGen first employs Grounded-SAM to localize task-relevant objects, and then queries the ConceptFactory database (Sun et al., 2025) to retrieve corresponding parameterized templates defined by primitives.

Formally, we represent each object part with a parameter tuple specifying its pose and primitive-specific attributes. For a scene with $N_{parts}$ object parts, the parameter list is defined as

$$P = \{(x_i, y_i, z_i, \alpha_i, \beta_i, \gamma_i, s_i^1, \ldots, s_i^M)\}_{i=1}^{N_{parts}}$$

where $(x_i, y_i, z_i)$ and $(\alpha_i, \beta_i, \gamma_i)$ represent the 3D position and orientation (e.g., Euler angles) of the $i$-th primitive in the global coordinate system, while $(s_i^1, \ldots, s_i^M)$ correspond to its shape-specific parameters (e.g., radius, height, width depending on primitive type).

In parallel, the RGB observation is processed by a visual encoder to extract features, which are passed into a lightweight parameter estimator to predict the entries of $P$. The predicted parameters are then used to instantiate the retrieved templates with atomic geometric primitives, yielding the complete structure map $\mathcal{M} = \{\mathcal{S}_{o_j}\}_{j=1}^K$. Through this end-to-end process, StructGen encodes both geometric and topological regularities, enabling scalable construction of structure maps that generalize across diverse objects and manipulation scenarios.

### 3.4 STRUCTTRANSFORMER

To incorporate the structure map produced by the StructGen module into the downstream control policy, we introduce *StructTransformer*, which extracts map features from the structure map. By reasoning directly over structural points, StructTransformer enables the policy to exploit object-centric and topological information without modifying the backbone policy architecture.

The architecture consists of two parallel streams. The first encodes the target pose, which specifies the desired robot–object spatial relation for the task. A lightweight MLP-based encoder processes the target pose into the pose features $f_p$ that provide task-specific conditioning signals. The second stream extracts map features from the structure map $\mathcal{M} = \{(\mathbf{p}_n, \ell_n, o_n)\}_{n=1}^{N_M}$, where each label $\ell_n$ indicates the type of atomic geometric primitive (AGP). A hierarchical encoding strategy is adopted: the structure map is first partitioned into groups by AGP labels, and each group is independently processed by a dedicated StructTransformer Block, yielding primitive-level representations. The outputs of these encoders, referred to as Primitive Tokens, capture structural knowledge specific to each primitive. These tokens are then passed into another StructTransformer Block, which reasons over inter-primitive relations and encodes topological dependencies across the object, producing the final global features $f_g$ of the structure map. Finally, the map features $f_M$ are obtained by concatenating $f_g$ with the pose features $f_p$.

Each StructTransformer Block is a transformer-based network tailored for structural point sets. It takes a set of 3D coordinates $(x_j, y_j, z_j)_{j=1}^{N_M}$ as input, applies two MLPs to generate point-wise and positional embeddings, computes global self-attention across the entire set, and aggregates the resulting features through attention pooling to form a compact token.

## 4 EXPERIMENTS

In Section 4.1, we present the simulated experiments on *MetaWorld*, *CALVIN*, and *Push-T*, detailing the setups and reporting the results. The robustness of StructPolicy under different visual changes including visual noise, camera viewpoint shifts, light intensity, and background color variations is evaluated in Section 4.2. In Section 4.3, we validate the effectiveness of the structure map representation and encoder through ablation studies, and further analyze the impact of the hyperparameter $N$ in atomic geometric primitives. The inference efficiency of StructPolicy is evaluated in Section 4.4. Finally, we provide additional results, including real-world experiments in Appendix A and extended analyses on *Push-T* in the Appendix B.1.

### 4.1 SIMULATED EXPERIMENTS

#### 4.1.1 EVALUATION ON METAWORLD

We evaluate StructPolicy on MetaWorld (Todorov et al., 2012), a MuJoCo-based benchmark with 50 classical robot manipulation tasks across four difficulty levels. Following (Jia et al., 2025), we select 15 representative tasks spanning easy (*button-press, drawer-open, reach, handle-pull, peg-unplug-side, lever-pull, dial-turn*), medium (*hammer, sweep-into, bin-picking, push-wall, box-close*), and hard/very hard (*assembly, hand-insert, shelf-place*). We adopt the Clip version of Lift3D (Jia et al., 2025) as the backbone of StructPolicy, which extends 2D foundation models to 3D manipulation via affordance masking and depth reconstruction. Training uses 30 demonstrations per task, 100 epochs

| Level | Easy | | | | | | | | Very Hard |
|-------|------|---|---|---|---|---|---|---|-----------|
| Task | Button-press | Drawer-open | Reach | Handle-pull | Peg-unplug-side | Lever-pull | Dial-turn | **Mean S.R.** | Shelf-place |
| Lift3D(Dinov2) | 100 | 100 | **80** | 100 | 96 | 76 | 100 | 93.1 | 28 |
| Lift3D(Clip) | 100 | 100 | 74 | 100 | **98** | 86 | 100 | 94.0 | 42 |
| **StructPolicy (ours)** | 100 | 100 | 76 | 100 | **98** | **96** | 100 | **95.7** | **58** |

| Level | Medium | | | | | | Hard | | All |
|-------|--------|---|---|---|---|---|------|---|-----|
| Task | Bin-picking | Box-close | Hammer | Push-wall | Sweep-into | **Mean S.R.** | Assembly | Hand-insert | **Mean S.R.** | **Mean S.R.** |
| Lift3D(Dinov2) | **100** | 92 | **100** | 40 | 80 | 82.4 | 100 | 76 | 88 | 84.5 |
| Lift3D(Clip) | 92 | 92 | 94 | 44 | 72 | 78.8 | 100 | 64 | 82 | 83.9 |
| **StructPolicy (ours)** | 96 | **100** | 98 | **62** | **88** | **88.8** | 100 | **86** | **93** | **90.5** |

Table 1: Comparison of manipulation success rates between StructPolicy and LIFT3D baselines. The table presents task-specific scores for each method, covering 15 tasks in Metaworld.

with rollouts every 10, and we report the maximum average success rate across tasks, evaluated from two camera viewpoints and averaged

As shown in Table 1, our StructPolicy consistently outperforms all baselines across all 15 tasks, including both CLIP-based and DINOv2-based Lift3D policies. Notably, StructPolicy not only achieves new state-of-the-art results but also surpasses the 90% success rate threshold, setting a new benchmark on these tasks to the best of our knowledge.

### 4.1.2 EVALUATION ON CALVIN

| Train→Test | Method | No. Instructions in a Row (1000 chains) | | | | | | |
|------------|--------|-----|-----|-----|-----|-----|-----------|------|
| | | 1 | 2 | 3 | 4 | 5 | **Avg. Len.** | **Rank** |
| D→D | **TaKSIE** | **90.4%** | **73.9%** | **61.7%** | **51.2%** | **40.8%** | **3.18** | **6** |
| | **HULC++** | **93.0%** | **79.0%** | **64.0%** | **52.0%** | **40.0%** | **3.30** | **5** |
| | **MDT** | **93.3%** | **82.4%** | **71.5%** | **60.9%** | **51.1%** | **3.59** | **4** |
| | **MDT-V** | **93.7%** | **83.2%** | **71.7%** | **60.5%** | **50.6%** | **3.72** | **3** |
| | **RoboUniView** | **96.2%** | **88.8%** | **77.6%** | **66.6%** | **56.3%** | **3.85** | **2** |
| | **StructPolicy (ours)** | **93.9%** | **85.3%** | **76.7%** | **68.9%** | **61.6%** | **3.86** | **1** |
| ABCD→D | **DeeR** | **98.2%** | **90.2%** | **82.1%** | **75.9%** | **67.0%** | **4.13** | **6** |
| | **GR-1** | **94.9%** | **89.6%** | **84.4%** | **78.9%** | **73.1%** | **4.21** | **5** |
| | **MoDE** | **97.1%** | **92.5%** | **87.9%** | **83.5%** | **77.9%** | **4.39** | **4** |
| | **MDT** | **97.5%** | **92.4%** | **87.1%** | **81.4%** | **74.8%** | **4.33** | **3** |
| | **MDT-V** | **98.8%** | **95.9%** | **91.2%** | **86.1%** | **79.4%** | **4.51** | **2** |
| | **StructPolicy (ours)** | **98.7%** | **96.2%** | **92.7%** | **88.7%** | **83.9%** | **4.60** | **1** |

Table 2: Performance comparison of various policies learned end-to-end on the CALVIN ABCD→D and D→D challenge within the CALVIN benchmark. We show the average rollout length to solve 5 instructions in a row (Avg. Len.) of 1000 chains.

We evaluate StructPolicy in the CALVIN challenge (Mees et al., 2022), which consists of four similar but distinct environments (A, B, C, D) featuring a multifunctional table and three blocks with varying shapes and sizes. Two standard settings are considered: **ABCD → D**, where the policy is trained on all four environments with 24 hours of uncurated teleoperated play data covering 34 tasks and tested on environment D, and **D → D**, where both training and evaluation are conducted on environment D with a smaller 6-hour dataset. We adopt MDT-V (Reuss et al., 2024) as the backbone of StructPolicy, a transformer-based diffusion policy with latent goal-conditioned state representations that ranks first on the CALVIN leaderboard. Evaluation follows the long-horizon benchmark, which contains 1000 natural language instruction chains, each requiring the robot to continuously solve 5 tasks. Agents receive a reward of 1 per successful instruction, with a maximum score of 5 per rollout.

As shown in Table 2, our proposed **StructPolicy** achieves new state-of-the-art performance on CALVIN Leaderboard under both evaluation settings. On the ABCD → D challenge, where the policy is trained on four environments and tested on a distinct one, StructPolicy achieves an average instruction chain length of **4.60**, surpassing MDT-V (**4.51**) and demonstrating strong **robustness** to

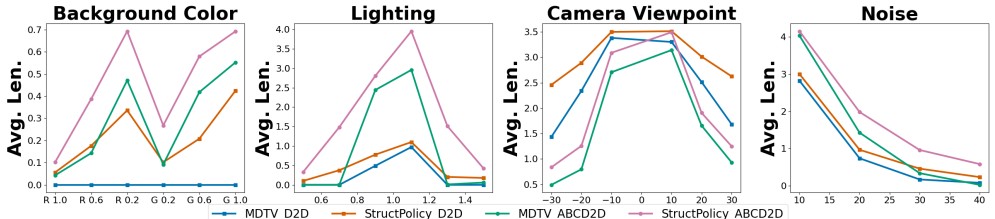

Figure 3: Illustrations of the visual changes used to evaluate policy robustness. Lighting: varying light intensity Background color: red and green backgrounds with different channel strengths. Camera viewpoint: horizontal rotations of the camera by *P 10°*, *P 30°* (positive direction) and *N 10°*, *N 30°* (negative direction). Noise: Gaussian noise added to images with different levels.

Figure 4: Quantitative results of policy performance under visual domain shifts in the CALVIN environment. StructPolicy consistently outperforms MDT-V across all perturbation types, including lighting, background color, camera viewpoint, and noise.

environment shifts. In the low-data D → D setting, it still achieves the best score of **3.86**, outperforming RoboUniView (**3.85**) and MDT-V (**3.72**).

## 4.2 ROBUSTNESS STUDIES

We further investigate the robustness of StructPolicy under domain shifts, using MDT-V and Lift3D as baselines. Specifically, we evaluate policy performance under four types of visual changes at test time: *1) Lighting* with intensity levels 0.5–1.5 relative to default; *2) Background color* with red or green channel values varied across 0.2, 0.6, and 1.0; *3) Camera viewpoint* with horizontal shifts of 10°, 20°, and 30°; and *4) Noise* with Gaussian perturbations of std = 10–40.

Illustrations of visual changes are shown in Figure 3, and the quantitative results in Figure 4 highlight StructPolicy's clear advantages over MDT-V across all settings. StructPolicy remains stable under strong background color changes, adapts to both darker and brighter lighting, and sustains robustness under camera rotations up to ±30°, while MDT-V degrades significantly. Even with heavy Gaussian noise (e.g., std = 40), StructPolicy maintains higher accuracy, underscoring its resilience to sensor imperfections. These results confirm that leveraging domain invariant structure maps enables StructPolicy to generalize more effectively under distribution shifts than baselines.

Additional robustness experiments with Lift3D are provided in Appendix C.2.

## 4.3 ABLATION STUDIES

### 4.3.1 STRUCTMAP REPRESENTATION AND ENCODER

As shown in Table 3, we conduct an ablation study on MetaWorld to evaluate the impact of the StructTransformer and the sparse object structure representation. We compare StructTransformer with three baselines—PointNet (PN), PointTransformer (PT), and DenseFusion (DF)—under both sparse structure and uniform point sampling. Results show that StructTransformer achieves the highest performance with sparse input, highlighting its ability to capture task-relevant structural features through global attention. Moreover, sparse representation consistently outperforms uniform

| | Map Encoder | | | | Object Structure | | Mean | | | Map Encoder | | | | Object Structure | | Mean |
|---|---|---|---|---|---|---|---|---|---|---|---|---|---|---|---|---|
| | ST | PT | DF | PN | Sparse | Uniform | | | | ST | PT | DF | PN | Sparse | Uniform | |
| Ex1 | - | - | - | ✓ | - | ✓ | **79.8** | Ex2 | | - | - | - | ✓ | ✓ | - | **80.3** |
| Ex3 | - | - | ✓ | - | - | ✓ | **83.2** | Ex4 | | - | - | ✓ | - | ✓ | - | **84.0** |
| Ex5 | - | ✓ | - | - | - | ✓ | **79.6** | Ex6 | | - | ✓ | - | - | ✓ | - | **80.8** |
| Ex7 | ✓ | - | - | - | - | ✓ | **80.5** | Ex8 | | ✓ | - | - | - | ✓ | - | **90.5** |

Table 3: Ablation study on structmap representation and encoder. In Map Encoder, ST means StructTransformer, PT means Point Transformer, DF means DenseFusion and PN means PointNet. In Object Structure, Sparse means the sparse object structure representation, Uniform means sampling points uniformly from objects' surface.

| Number $N$ | 3 | 6 | 8 | 12 | 18 | 24 | 36 |
|---|---|---|---|---|---|---|---|
| Corner 1 | 89.1 | 89.6 | **90.1** | 90.1 | 89.6 | 89.3 | 89.1 |
| Corner 2 | 88.3 | 88.8 | **90.4** | 90.1 | 89.6 | 89.1 | 88.8 |
| **Mean S.R.** | 88.7 | 89.2 | **90.3** | 90.1 | 89.6 | 89.2 | 88.9 |

(a)

| Method | Lift3D(CLIP) | StructPolicy |
|---|---|---|
| Average Inference Time (ms) ↓ | 67.21 | 70.20(+2.99) |

(b)

Table 4: Results on MetaWorld tasks: (a) Ablation study on $N$ in sparse object representations. (b) Inference speed comparison between Lift3D and StructPolicy.

sampling across all encoders, confirming the importance of emphasizing semantic and topological keypoints. Together, StructTransformer and sparse representation yield the best success rate, demonstrating their complementary roles in robust policy learning.

### 4.3.2 OBJECT STRUCTURE REPRESENTATION

We further conduct an ablation study to investigate how the choice of hyperparameter $N$ affects performance. Specifically, we vary $N$ across a range of values: 3, 6, 8, 12, 18, 24, and 36. As shown in Table 4 (a), the performance improves as $N$ increases, reaching a peak at $N = 12$ and $N = 8$ under the *corner 1* camera view, and peaking at $N = 8$ under the *corner 2* view. Beyond these values, performance begins to plateau or slightly decline. This trend suggests a trade-off in the number of sampled structural points: when $N$ is too small, the structure map lacks sufficient detail to represent object geometry, reducing its utility for policy learning. Conversely, when $N$ becomes too large, the increased density introduces redundancy and may blur the salient topological cues, weakening the structural guidance effect.

### 4.4 INFERENCE SPEED

To assess the inference cost of our approach, we report inference speeds averaged over 15 Meta-World tasks, as summarized in Table 4 (b). StructPolicy exhibits a negligible overhead of only 2.99 milliseconds, compared to the baseline Lift3D (CLIP) model, demonstrating that the introduced structured representation incurs minimal runtime cost while enabling improved performance.

## 5 CONCLUSION

In this work, we present StructPolicy, a lightweight and effective module for Imitation Learning that explicitly incorporates object-level structural knowledge into policy learning. By constructing a stable and task-relevant structure map, our method helps the robot focus on topological cues critical for manipulation, reducing its reliance on raw visual details that often cause brittleness under distribution shifts. To build the structure map, we develop a general and sparse object representation scheme using a set of parameterized atomic geometries, allowing flexible composition for a wide range of manipulation tasks. We further propose StructTransformer, a global-attention-based encoder that extracts informative structural features from the structure map to guide action prediction. Extensive experiments across 50 tasks, 3 IL baselines, and 3 simulated environments show that StructPolicy consistently improves performance while introducing minimal inference overhead.

## ETHICS STATEMENT

All datasets used in this work are publicly available and derived from open-source simulation environments. We ensure that no proprietary or sensitive data is involved. For the real-world experiments, we commit to releasing the collected datasets to the community to promote transparency, reproducibility, and fair comparison. Our work does not involve human subjects, sensitive personal information, or applications with foreseeable harmful impacts.

## REPRODUCIBILITY STATEMENT

We are committed to ensuring the reproducibility of our work. The implementation details, including training configurations and evaluation scripts, are provided in the supplementary material (see the *Code* section). All datasets used in this study are publicly available: CALVIN, MetaWorld, and Push-T datasets have been released to facilitate reproducibility and fair comparison.

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

APPENDIX

## A    REAL-WORLD EXPERIMENTS

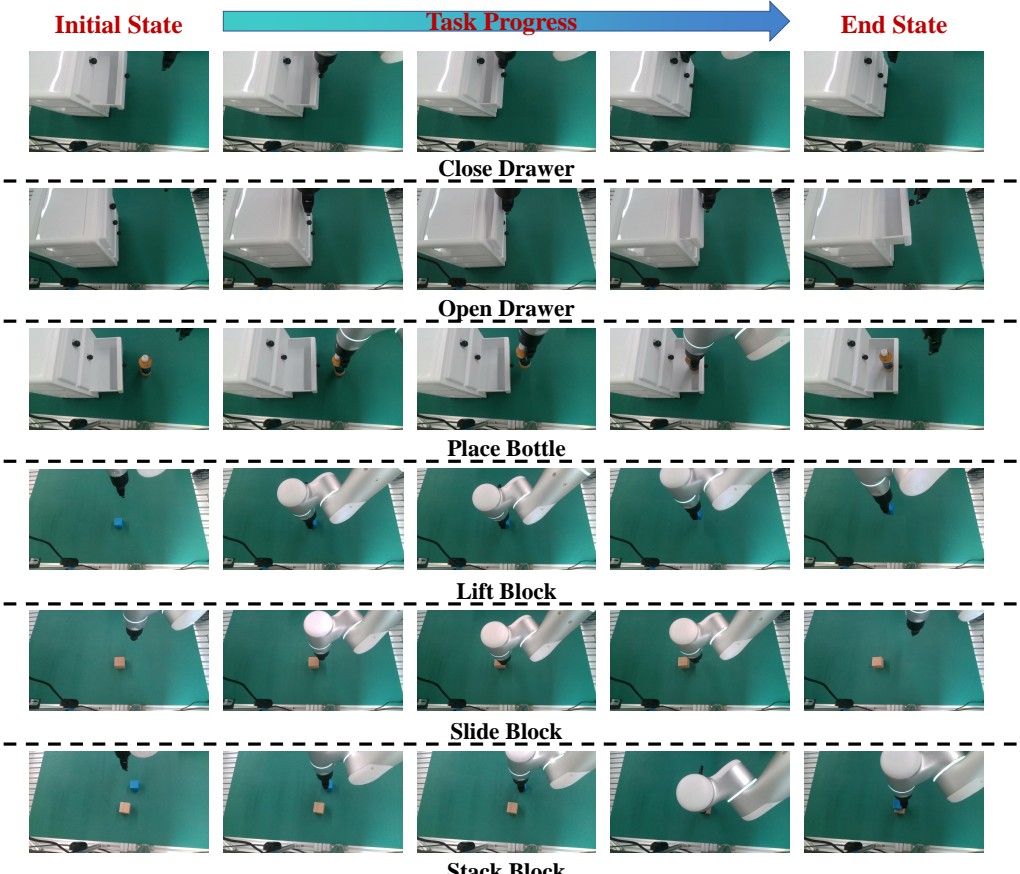

Figure 5: Visualization of real-world experiments across six tasks: close drawer, open drawer, place bottle, lift block, slide block, and stack block.

### A.1    IMPLEMENTATION DETAILS

We deploy our method on a Flexiv Rizon 4 robotic arm with a 7-DoF configuration and a Flexiv-GN01 parallel gripper, operating at 30 Hz. Visual inputs are captured using an NVIDIA RealSense L515 RGB-D camera. The experimental setup involves two wooden blocks, a storage drawer, and a bottle.

We adopt Lift3D as the backbone policy, and adapt its output space from simulation actions to an 8-dimensional target pose representation, consisting of the 3D end-effector position, a 4D quaternion for orientation, and a 1D gripper width. Collected trajectories are normalized to enhance training stability. At test time, target poses are executed using the `robot.ExecutePrimitive` API from Flexiv RDK, while gripper commands are controlled via `gripper.Move`.

Since Lift3D operates on point cloud inputs, RGB-D images are first converted to point clouds using Open3D, followed by farthest-point sampling to 1024 points before being fed into the model.

### A.2    TASKS AND EVALUATION METRICS

We evaluate our approach on six representative real-world manipulation tasks designed to span a wide range of skills and interaction modalities. In the **Lift Block** task, the robot must accurately

grasp and lift a small block from the tabletop, testing the ability to combine perception and stable grasp control. The **Slide Block** task requires pushing a block to a designated target location, demanding precise end-effector positioning and force modulation. In the **Stack Block** task, the robot must place one block on top of another, which introduces the challenge of accurate object alignment and vertical placement.

For articulated-object interaction, we design two drawer-related tasks. In the **Open Drawer** task, the robot pulls a closed drawer outward until it reaches a fully open state, while in the **Close Drawer** task, it pushes the drawer back to its original closed position. Both tasks require coordinated motion along constrained trajectories and robustness to contact dynamics. Finally, the **Place Bottle** task evaluates fine-grained manipulation and spatial reasoning skills by requiring the robot to grasp a near-cylindrical bottle and place it precisely into the bottom drawer of a small cabinet without collision.

These six tasks collectively cover a broad spectrum of manipulation capabilities, including pick-and-place, pushing, stacking, articulated-object control, and precise placement. Each policy is evaluated over multiple trials under diverse initial conditions, and success rates are reported as the primary metric. A trial is considered successful if the robot completes the task goal within a fixed time horizon without failure modes such as dropping, misalignment, or incomplete motion execution.

## A.3 RESULTS

| Method | Lift Block | Slide Block | Stack Block | Open Drawer | Close Drawer | Place Bottle | Mean |
|---|---|---|---|---|---|---|---|
| Lift3D | 90 | 85 | 35 | 60 | 75 | 60 | 67.5 |
| **StructPolicy** | **95** | **95** | **55** | **80** | **80** | **95** | **72.5** |

Table 5: **Quantitative results on real-world robot experiments.** Each policy is evaluated over 20 trials per task under diverse initial conditions.

Table 5 reports the quantitative results of real-world experiments across six representative manipulation tasks, with qualitative demonstrations shown in Figure 5. Compared with the baseline *Lift3D*, StructPolicy achieves substantial performance gains on all tasks, with improvements particularly notable on more challenging scenarios such as *Stack Block* (+20%) and *Place Bottle* (+35%). On average, StructPolicy improves the overall success rate from 67.5% to 82.5%, demonstrating consistent advantages across diverse skill categories, including pick-and-place, pushing, stacking, articulated-object manipulation, and precise placement.

Qualitative observations further confirm that StructPolicy produces more stable and precise behaviors, such as reliably aligning the gripper when placing the bottle into the drawer and maintaining block stability during stacking. These results indicate that the structural representations learned in simulation can effectively transfer to the real world, yielding robust generalization and improved manipulation capabilities.

## B ADDITIONAL QUANTITATIVE EXPERIMENTS

### B.1 EVALUATION ON PUSH-T

**Environment** We utilize the variant of this task provided by (Chi et al., 2024) which involves pushing a T-shaped block(gray) to a designated target position (red) using a circular end effector (blue). Variation is introduced through randomized initial positions of the T block and the end-effector. We report success rate which corresponds to the T-block covering at least 0.95 of the target.

**Baseline** We choose the IMLE policy (Rana et al., 2025) as the backbone of StructPolicy. IMLE introduces Implicit Maximum Likelihood Estimation into robot manipulation and excels in multimodal tasks like *push-t*, offering faster inference than diffusion and flow-matching methods by learning directly from each trajectory.

| Method | 100% Dataset | 20 Demos |
|---|---|---|
| **StructPolicy (ours)** | **0.68/0.63** | **0.12/0.08** |
| IMLE Policy | 0.59/0.54 | 0.10/0.07 |
| IMLE Policy (w/out consistency) | 0.56/0.53 | 0.05/0.03 |
| Diffusion Policy | 0.57/0.52 | 0.03/0.03 |
| Flow Policy (Zhang et al., 2024) | 0.36/0.34 | 0.012/0.00 |

Table 6: Push-T Success rates are shown as (max success rate) / (average of last 3 checkpoints), averaged over 3 seeds and 50 env initializations.

**Evaluation Metrics**  We evaluate all methods on 100% dataset and 20 demonstrations, respectively. For each method, we evaluate across 3 training seeds and 50 different environment initial conditions.

**Results**  As shown in Table 6, StructPolicy exceeds IMLE policy with nearly **10%** absolute percent in 100% dataset. StructPolicy also shows strong data efficiency when training on 20 demonstrations, exceeding IMLE policy by **20%**.

## B.2 IMPLEMENTATION DETAILS

In our experiments, we use a lightweight MLP as the Parameter Estimator, consisting of a three-layer Linear–ReLU stack with a hidden dimension of 256. For template retrieval in the ConceptFactory database, objects extracted by Grounded-SAM are matched against assets in the database via similarity computation to obtain the closest template. The StructTransformer is built with point-wise feature extraction and positional embedding using an MLP (Linear–ReLU–Linear), followed by Transformer blocks for attention computation (Zhao et al., 2021). The final structural representation is obtained through attention pooling. The optimizer and learning rate scheduler are kept consistent with the backbone, and training is performed end-to-end with a unified loss. On the CALVIN benchmark, we train with the CALVIN dataset (Mees et al., 2022); in MetaWorld, we use procedurally generated data with 30 demonstrations per task (Jia et al., 2025); and on Push-T, we follow IMLE and adopt the same training dataset (Rana et al., 2025). Code is provided in the Supplementary Material, including modules for integrating StructPolicy into both MDT-V (`structpolicy_mdt`) and Lift3D (`structpolicy_lift3d`).

## B.3 FEW-SHOT LEARNING

| Level | Easy | | | | | | | | Very Hard |
|---|---|---|---|---|---|---|---|---|---|
| **Task** | Button-press | Drawer-open | Reach | Handle-pull | Peg-unplug-side | Lever-pull | Dial-turn | **Mean S.R.** | Shelf-place |
| Lift3D(Clip) | 100 | 94 | 36 | 82 | 90 | 70 | 100 | 81.7 | 16 |
| **StructPolicy (ours)** | 100 | **100** | **62** | **100** | **92** | **86** | 100 | **91.4** | **28** |

| Level | Medium | | | | | | Hard | | All |
|---|---|---|---|---|---|---|---|---|---|
| **Task** | Bin-picking | Box-close | Hammer | Push-wall | Sweep-into | **Mean S.R.** | Assembly | Hand-insert | **Mean S.R.** | **Mean S.R.** |
| Lift3D(Clip) | **72** | 84 | **66** | 44 | 56 | 64.4 | 72 | 52 | 62 | 68.9 |
| **StructPolicy (ours)** | 68 | **92** | 64 | **50** | **64** | **67.6** | **74** | **58** | **66** | **75.9** |

Table 7: Success rates of StructPolicy vs. Lift3D under limited demonstrations. Comparison of performance across 15 MetaWorld tasks using only 10 demos per task.

We further evaluate the data efficiency of StructPolicy by conducting experiments with reduced demonstration data on the 15 tasks from MetaWorld (Yu et al., 2020). While the main experiment on MetaWorld uses 30 demonstrations per task (25 for training, 5 for validation), here we reduce the number to only 10 demonstrations per task, with 8 used for training and 2 for validation. All other experimental settings remain consistent with the setup of experiments in the paper. Detailed results are reported in Table 7.

On the 7 easy tasks, StructPolicy maintains strong performance, achieving a mean success rate of 91.4% compared to Lift3D (Jia et al., 2025)'s 81.7%. Notably, in tasks like *Drawer-open*, *Handle-*

*pull*, and *Lever-pull*, StructPolicy achieves near-perfect success, showing almost no degradation compared to the full-data setting. Even on the challenging *Reach* task, performance improves significantly from 36% to 62%.

For medium-level tasks, StructPolicy achieves a higher average success rate of 67.6% versus 64.4% for Lift3D. It surpasses the baseline in 3 out of 5 tasks, especially in *Box-close* (+8%) and *Sweep-into* (+8%).

In the 2 hard tasks, StructPolicy also outperforms Lift3D, scoring 66% vs. 62%. For example, in *Hand-insert*, our method improves success rate by 6 percentage points (58% vs. 52%).

On the most difficult very hard task, *Shelf-place*, StructPolicy yields a substantial gain (28% vs. 16%) highlighting its robustness in complex, contact-rich settings.

Overall, StructPolicy delivers a 7% absolute improvement in mean success rate (75.9% vs. 68.9%) over the baseline across all 15 tasks, demonstrating not only strong general performance but also excellent data efficiency. These results confirm that guiding robots to focus on structural knowledge enables more efficient skill acquisition even in few-shot scenarios.

### B.4 STATISTICAL TESTS

To assess the statistical significance of our performance gain on MetaWorld, we conduct a Wilcoxon signed-rank test comparing StructPolicy and the Lift3D (Clip) baseline across 15 tasks. The resulting p-value is 0.0076 ($p < 0.01$) with test statistic $T = 0 < 1$, indicating a statistically significant difference between the two methods. Notably, the test statistic is 0, reflecting that StructPolicy consistently outperforms Lift3D on all tasks, which further confirms the robustness and effectiveness of our approach on this benchmark.

### B.5 MODULE EFFICIENCY AND ADAPTABILITY

StructPolicy introduces only a modest overhead of 1.73MB in model parameters compared to the baseline backbone. Moreover, once the atomic geometry-based object structures are defined, adapting StructPolicy to new IL models typically requires fewer than 100 lines of additional code, along with minor adjustments to the parameters of StructGen and StructTransformer. These aspects highlight that StructPolicy is a lightweight and easily transferable module, making it practical for broad adoption in diverse robotic manipulation scenarios.

## C ABLATION STUDIES DETAILS

### C.1 DETAIL SCORES OF ABLATION STUDY

Table 8 presents detailed success rates for each task in the MetaWorld benchmark, evaluating different combinations of map encoders and object structure representations. The experiments cover 15 manipulation tasks, and each reported score is the average success rate across two camera views: *corner1* and *corner2*.

Among all settings, StructTransformer with the proposed sparse representation (Ex8) consistently achieves the highest success rates across almost all tasks and difficulty levels, demonstrating the effectiveness of both components. Notably, on challenging tasks such as *Shelf-place* and *Hand-insert*, Ex8 outperforms other configurations by a significant margin, indicating its superior structural reasoning capabilities. Overall, the results validate that combining a well-designed sparse structure representation with a topology-aware encoder like StructTransformer is key to robust policy learning.

### C.2 ROBUSTNESS STUDIES ON METAWORLD

To further validate the robustness of StructPolicy in MetaWorld, we conduct additional experiments focusing on two challenging types of domain shifts: **1) Camera Viewpoints:** We change the camera viewpoint by shifting the camera vertically and horizontally by 10°, 20°, and 30°. **2) Random Noise:**

| Level | Easy | | | | | | | | Very Hard |
|---|---|---|---|---|---|---|---|---|---|
| Task | Button-press | Drawer-open | Reach | Handle-pull | Peg-unplug-side | Lever-pull | Dial-turn | **Mean S.R.** | Shelf-place |
| Ex1(PN+Uniform) | 100 | 100 | 72 | 96 | 96 | 92 | 100 | 93.7 | 24 |
| Ex2(PN+Sparse) | 100 | 100 | 72 | 100 | 84 | 86 | 100 | 91.7 | 36 |
| Ex3(DF+Uniform) | 100 | 100 | 72 | 100 | 94 | 88 | 98 | 93.1 | 46 |
| Ex4(DF+Sparse) | 100 | 100 | 72 | 98 | 96 | 90 | 98 | 93.4 | 52 |
| Ex5(PT+Uniform) | 100 | 100 | 72 | 100 | 92 | 94 | 100 | 94.0 | 38 |
| Ex6(PT+Sparse) | 98 | 100 | 72 | 92 | 84 | 92 | 100 | 91.1 | 42 |
| Ex7(ST+Uniform | 100 | 100 | 72 | 100 | 98 | 86 | 100 | 93.7 | 46 |
| Ex8(ST+Sparse) | 100 | 100 | 76 | 100 | 98 | 96 | 100 | 95.7 | 58 |

| Level | Medium | | | | | | Hard | | All |
|---|---|---|---|---|---|---|---|---|---|
| Task | Bin-picking | Box-close | Hammer | Push-wall | Sweep-into | **Mean S.R.** | Assembly | Hand-insert | **Mean S.R.** | **Mean S.R.** |
| Ex1(PN+Uniform) | 84 | 94 | 60 | 44 | 82 | 72.8 | 96 | 56 | 76 | 79.8 |
| Ex2(PN+Sparse) | 76 | 86 | 80 | 42 | 78 | 72.4 | 94 | 70 | 82 | 80.3 |
| Ex3(DF+Uniform) | 88 | 90 | 88 | 34 | 76 | 75.2 | 96 | 78 | 87 | 83.2 |
| Ex4(DF+Sparse) | 86 | 90 | 94 | 36 | 78 | 76.8 | 98 | 72 | 85 | 84.0 |
| Ex5(PT+Uniform) | 90 | 94 | 48 | 40 | 66 | 67.6 | 98 | 62 | 80 | 79.6 |
| Ex6(PT+Sparse) | 94 | 96 | 70 | 42 | 72 | 74.8 | 96 | 62 | 79 | 80.8 |
| Ex7(ST+Uniform | 90 | 96 | 76 | 34 | 58 | 70.8 | 94 | 58 | 76 | 80.5 |
| Ex8(ST+Sparse) | 96 | 100 | 98 | 62 | 88 | 88.8 | 100 | 86 | 93 | 90.5 |

Table 8: Detailed per-task success rates of ablation study. This table provides full results for each task to complement the ablation study in the main paper.ST means StructTransformer, PT means Point Transformer, DF means DenseFusion and PN means PointNet. Sparse means the sparse object structure representation, Uniform means sampling points uniformly from objects' surface.

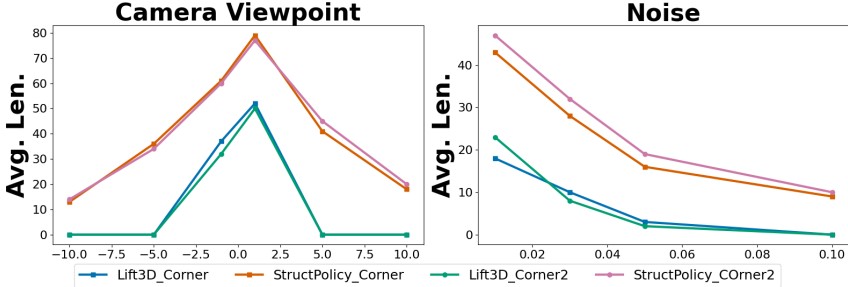

Figure 6: Quantitative results of policy performance under visual domain shifts in the MetaWorld environment. StructPolicy consistently outperforms Lift3D across all perturbation types, including camera viewpoint and noise.

We inject Gaussian noise into the rendered images, with the standard deviation set to 10, 20, 30, and 40, to simulate sensor imperfections.

Since Lift3D encodes only 3D point positions without color features, we do not consider lighting or background variations and restrict our evaluation to viewpoint shifts and image noise. As shown in Figure 6, StructPolicy demonstrates significantly higher robustness across both perturbations, maintaining stable performance even when visual changes increase. In contrast, Lift3D suffers from substantial degradation, reflecting its reliance on perspective-specific cues and limited geometric abstraction. These results reinforce the advantage of leveraging structure maps in enabling viewpoint-invariant and noise-resilient policy learning.

# D  STRUCTURE OF ATOMIC GEOMETRIES

Here we provide detailed structure for atomic geometries shown in Figure 7:

- **Cuboid:** We divide each rectangular face into two triangles, and extract the resulting triangle vertices as structure points. This results in 12 triangles and 8 unique corner points.

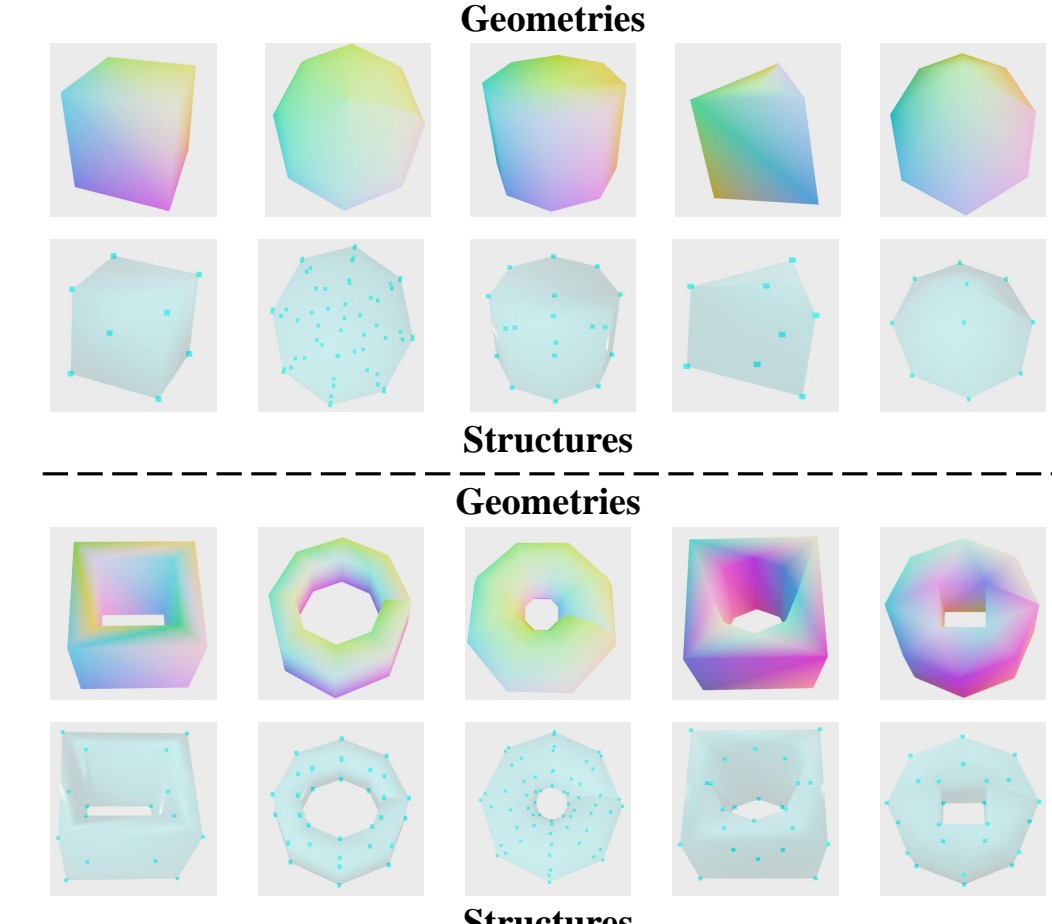

**Geometries**

**Structures**

**Geometries**

**Structures**

Figure 7: Atomic geometries structure with $N = 8$.

- **Sphere:** We generate $N$ longitudinal circular arcs and sample points uniformly along each arc using Uniform Circular Sampling. We ensure an even spread of points across the sphere surface by varying latitude.

- **Cylinder:** We sample $N$ points uniformly along multiple vertical arcs (parallel to the height axis) and horizontal arcs (on the top and bottom circular faces). Each arc lies on a cross-sectional plane.

- **Triangular Prism:** The prism is decomposed into flat polygonal surfaces (two triangular bases and three rectangles). Each surface is divided into triangles, and the triangle vertices are extracted as structure points.

- **Cone:** We sample points along vertical arc sections from the apex to the circular base using Uniform Circular Sampling, capturing both height and radial curvature. Base circles are also sampled uniformly.

- **Rectangular Ring:** Treated as a hollow box frame with six outer and six inner faces. Each face is triangulated, and all triangle vertices are included as structure points, preserving the hollow rectangular topology.

- **Ring:** Similar to a thin torus, we sample multiple radial arcs across the ring body. Each arc lies in a different plane, covering the circular shape of the ring with uniformly spaced points.

- **Torus:** We extract both major and minor circular arcs (along the ring and cross-section respectively), and sample points using two nested circular sampling loops, ensuring full 3D coverage of the toroidal surface.

- **Box-Cylinder-Ring (Compound Primitive):** This shape is composed of three parts: a cuboid base, a vertical cylinder, and a surrounding ring. We apply the decomposition strategy to the box, circular sampling to the cylinder and ring, and then merge all structure points into one composite set.

- **Cylinder-Box-Ring (Compound Primitive):** Similar to the above, but the base is a cylinder with a cuboid on top and a ring around. Each component is handled using its respective strategy and then aggregated to form the full structural representation.

# E QUALITATIVE EXPERIMENTS

## E.1 CAMERA VIEWPOINT SHIFT ON METAWORLD

We train both Lift3D and StructPolicy under the *Corner1* camera view and evaluate them under a novel view, *Corner2*, as illustrated in Fig. 8. Similarly, when training under *Corner2* and testing under *Corner1*, we observe consistent trends. The results show that StructPolicy consistently completes the tasks even under unseen viewpoints, demonstrating its ability to abstract object-level structural information from observations. Since the structure map remains invariant across camera perspectives, the robot can rely on it to perform accurately regardless of view changes.

In contrast, the baseline policy fails to generalize to the new view, exhibiting erratic behaviors due to overfitting to the training perspective. This highlights the robustness of StructPolicy and its capacity for cross-view generalization through structured perception.

For example, in the *dial-turn* task, although the Lift3D policy roughly identifies the location of the dial under the novel *Corner2* viewpoint, it fails to generate an accurate movement trajectory. The robot deviates from the dial's correct location, moving off-target due to its reliance on perspective-specific visual cues seen during training. This results in task failure despite a seemingly correct initial localization.

In contrast, StructPolicy precisely locates the dial and executes the correct rotation action without deviation. This success can be attributed to its structure-aware perception: by reasoning over a viewpoint-invariant structure map, StructPolicy isolates the task-relevant object geometry from camera-dependent appearance. This example further illustrates the robustness of StructPolicy in handling cross-viewpoint scenarios where traditional visual imitation policies tend to fail.

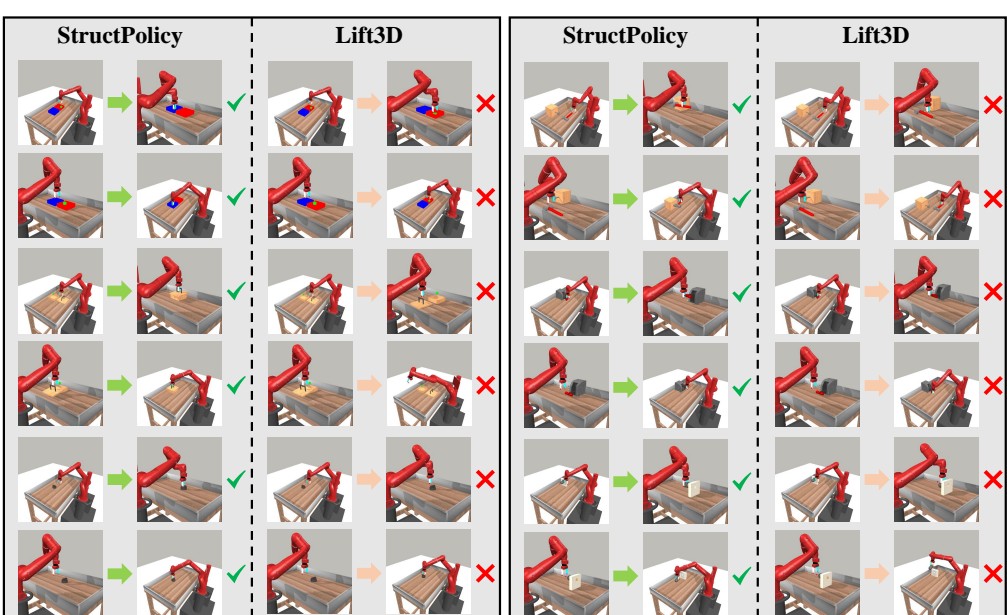

Figure 8: Visualization of camera viewpoint shifts in MetaWorld.

## E.2 Lighting Conditions on CALVIN Benchmark

We conduct qualitative experiments on the CALVIN benchmark (Mees et al., 2022) to investigate the sensitivity of StructPolicy to changes in lighting conditions. Specifically, we evaluate policies under both increased and decreased brightness levels. As shown in Figure 9, the baseline model MDT-V (Reuss et al., 2024) suffers significant performance degradation under both settings, failing to complete the task due to its reliance on raw visual features. In contrast, StructPolicy remains robust, successfully completing the task under both lighting variations.

During testing under lighting conditions that differ significantly from training, specifically, increased or decreased brightness, we observe a clear performance gap between StructPolicy and the MDT-V baseline.

The MDT-V policy, which relies heavily on raw visual inputs, exhibits severe degradation in performance when exposed to unseen lighting variations. In the *lift blue block* task, for example, the policy fails to localize the object precisely, with the end-effector stopping less than 1cm away from the block. This misalignment results in consistent task failure, highlighting the brittleness of perception-to-action mapping that depends on low-level visual features.

In contrast, StructPolicy maintains robust performance across both lighting shifts. While minor inaccuracies may occur compared to its original performance under normal illumination, the policy still completes the majority of tasks successfully. This robustness can be attributed to the guidance of the **structure map**, which encodes high-level, object-centric geometric information that remains invariant to lighting changes. Since the structure map abstracts away irrelevant pixel-level noise and emphasizes stable object structure, the learned policy becomes less sensitive to appearance-based perturbations.

These results demonstrate that StructPolicy enables the robot to learn manipulation-relevant structure representations, leading to improved robustness in dynamic or imperfect visual environments where conventional visual policies fail.

## LLM Usage

We primarily employed large language models (LLMs) for polishing the manuscript, aiming to improve fluency and clarity without altering the original meaning. All LLM-generated content was carefully reviewed and verified by the authors, who take full responsibility for the final text. All experiments, results, and conclusions were independently conducted by the authors without assistance from LLMs.

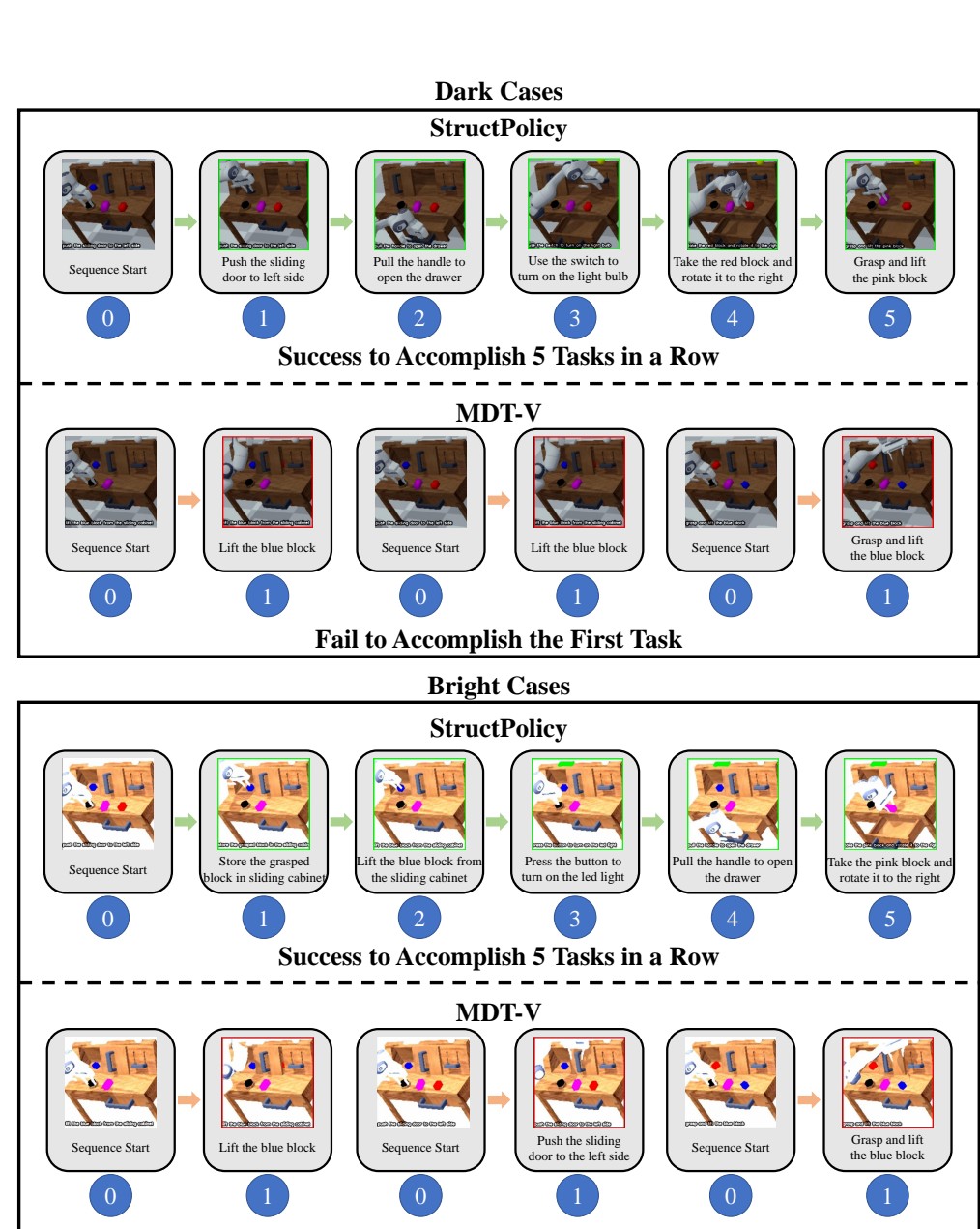

Figure 9: Light conditions on CALVIN Benchmark.

