# OpenReview forum: "StructPolicy: Robust Imitation Learning Policy Guided by Structure Map"
_ICLR.cc/2026/Conference — ICLR 2026 Conference Withdrawn Submission_

### Official Review · Reviewer_H3Xi · 2025-10-17

**Soundness:** 2
**Presentation:** 2
**Contribution:** 2
**Rating:** 2
**Confidence:** 4

**Summary:**

StructPolicy proposes a module for enhancing image-based Imitiation Learning (IL) policy performance and robustness to various visual perturbations. The module constructs representations of scenes based on RGB images and corresponding natural language task descriptions, which aim to capture task-relevant and visual-domain-agnostic features. These representations, named Structure Maps, are constructed using an elaborate pipeline with various components including pretrained segmentation models and an object library database. The authors demonstrate that training with the resulting representation enhances various IL policies on simulated and real-world robotic manipulation tasks.

**Strengths:**

Overview:
- The approach is modular and can be compatible with various IL policies.
- The overall representation extraction pipeline seems novel.
- Includes real-world experiments.
- Detailed figures of high-level pipeline.

**Weaknesses:**

**Overview**
- Complex representation extraction method with many moving parts.
- Method description lacks details for understanding the approach.
- Method does not result in significant empirical gains in performance.
- Representation claims to be “domain-invariant” but performance is significantly affected by visual perturbations (Figure 4, 6).
- Limitations are not discussed.

**Representation Pipeline**

The method section lacks details for the reader to fully understand and assess the approach:
- “StructGen first employs Grounded-SAM to localize task-relevant objects” (lines 264-265). How are the task-relevant objects specified exactly? What are the assumptions on the task prompt? Would this approach work if the task prompt would be in the form “move *all* objects to…”? Also, Grounded SAM is not cited in the paper if I am not mistaken.
- “then queries the ConceptFactory database” (line 265). I do not think a citation is sufficient here since this is a critical part of the pipeline. The paper should be relatively self-contained and provide details about how exactly a 2D image is converted to a set of 3D object parts as well as how the database search is performed. These should be provided in the Appendix and a detailed example from one of the domains in the experiment section would help the reader truly understand this process.
- “the RGB observation is processed … a lightweight parameter estimator to predict the entries of P” (lines 275-276). What is the training signal for the parameter estimator?
- “applies two MLPs to generate point-wise and positional embeddings” (lines 299-300). The difference between the point-wise and positional embeddings is not clear. Also how they are incorporated in each representation is not specified (concatenated, additive, etc.).

Given the above lack of detail it is hard to assess the robustness of this pipeline which relies on many moving parts. Some questions that arise include: How are objects that are not part of the task but may interfere with performing it (e.g., an obstacle) treated by this pipeline? How rich is the database and what is the price of a misidentification in terms of downstream performance? What is the accuracy of the predicted entries of P and how does it affect performance?
My main concern is that this pipeline may be brittle and hard to work with.
Conducting experiments that study the quality of intermediate pipeline outputs could shed light on this matter. Discussing and studying the limitations induced by each adopted component and by the overall approach is very important in my opinion.

**Object-level Reasoning**

Can the authors please explain why they believe their method facilitates object-level reasoning?
As I see it, the “reasoning” aspect is captured by the policy which receives an aggregated input which no longer corresponds to distinct objects or object parts.

**Experiments**

- The choice of baseline for the MetaWorld evaluation is not motivated. The performance in most of the tasks seem to be saturated. I believe choosing a baseline method that performs worse on these tasks would actually be a better fit because there would be a larger margin for improvement with your proposed module.
- The improvements on the CALVIN tasks without perturbation seem marginal to me. If this is not the case, the authors should clarify why this is not true.
- The authors claim their representation is “domain invariant” but the results in Figure 4 and 6 do not support this claim. Drops in performance are significant in many cases. Could this be related to the fact that a standard visual encoder is used to estimate the parameters of the AGPs which is not robust to the visual perturbations?
- Push-T is a 2D environment so I am not sure why your method is applicable in this setting. You claim to have results on PushT in the main text Section 4.1 but it is only presented in the Appendix.
- Real-world Experiments: I think these should be in the main text. I am particularly interested in details and visualizations of the intermediate pipeline stages for these experiments. Why were perturbation experiments not performed? I would think that real-world deployment is the main motivation for improving robustness to imperfect and changing visual conditions.

**Misc**

- Can the authors provide inference speed results for the other environments including the real-world experiments?
- Citations for the ablation baselines are missing.
- Standard deviation should be provided for the success rates.
- I suggest highlighting the best results in the table as opposed to all results for visual clarity.
- Every section of the appendix should be referenced from the main text in the relevant sections.

**Questions:**

See weaknesses.

---

### Official Review · Reviewer_8T7H · 2025-10-22

**Soundness:** 4
**Presentation:** 2
**Contribution:** 3
**Rating:** 8
**Confidence:** 4

**Summary:**

The paper proposes StructPolicy, a method to overcome visual distractions by constructing a task-relevant “Structure Map” - an object representation with geometric primitives that is integrated in the policy learning for imitation learning. Furthermore, to account for this representation, a modified transformer, StructTransformer, that applies hierarchical attention is proposed as the policy architecture. The approach can be integrated within several models and is evaluated on various environments and tasks.

**Strengths:**

* Novelty: my overall impression is that the method is a novel contribution and I find it highly relevant to the representation learning for decision making community.
* Comprehensive experimental suite.
* Strong performance compared to baselines.
* The method demonstrates better robustness to perturbations.
* Real-world experiment.
* A promise for open-source code.

**Weaknesses:**

* Reliance on supervision: one of the core components of the methods relies on a supervised segmentation and grounding model, which might be sensitive to out-of-distribution and/or unseen objects without any fine-tuning or adaptation. I find this a limitation of the model that should be discussed more explicitly.
* Limited to objects that can in fact be composed with the geometric primates in the ConceptFactory.
* Sections 3.2-3.3 clarity: I found the structure map generation process unclear. If this part builds upon processes described in previous work, I would suggest making the paper self-contained (at least for that part) and add a detailed description of this process (even if it ends up in the appendix). Perhaps a code block (e.g., torch/numpy/jax-style code block or just a pseudocode algorithm).
* No discussion of limitations (aside for the additional overhead). When would the method fail? What are the underlying assumptions required to construct the StructMap?


**Minor**

* Usage of “\cite”: seems like in a lot of places (e.g., Related Work), “\cite” or “\citep” is used instead of “\citet”.
* Standard deviations are missing from the tables.
* Section 4.3.2 - please remind the reader the meaning of $N$, as there are multiple notations using $N$ in the paper.

**Questions:**

* Can the authors visualize the StructMap for the real-world experiment?
* Can the authors please clarify my concerns raised under Weaknesses?

---

### Official Review · Reviewer_QqHY · 2025-10-29

**Soundness:** 3
**Presentation:** 3
**Contribution:** 3
**Rating:** 4
**Confidence:** 5

**Summary:**

This paper proposes StructPolicy, a module designed to improve the robustness of imitation learning for robotic manipulation. The key idea is to replace direct reliance on raw visual inputs, which are sensitive to variations in appearance, with a structure map that captures the geometric composition and spatial relationships of task-relevant objects. The system first constructs this map using a set of parameterized geometric primitives, and then encodes it using a transformer-based model to provide structural guidance to the control policy. By focusing on object-level topology rather than pixel-level detail, StructPolicy enables policies to generalize more reliably across changes in lighting, background, and camera viewpoint. Experiments on MetaWorld, CALVIN, Push-T, and real robot tasks demonstrate consistent improvements in success rates and robustness.

**Strengths:**

The paper is well-motivated and clearly addresses a real limitation in imitation learning: policies often overfit to visual appearance and fail when lighting, background, or camera viewpoint changes. The proposed structure map offers a clean and interpretable way to represent task-relevant geometry, helping the policy focus on spatial relationships rather than pixel details. The method is also practical, since StructPolicy is designed as a small plug-in module that can be added to existing IL models without major architectural changes. The writing is generally clear, and the overall pipeline is easy to follow. Most notably, most simulation experiments are thorough and convincing, covering multiple benchmarks and demonstrating consistent improvements in robustness and success rates across many tasks and visual perturbations. These simulation results strongly support the core claims, even though the real-robot demonstrations are more limited in scope.

**Weaknesses:**

A central limitation of the paper lies in the strong underlying hypothesis that high-level object structure alone is sufficient for manipulation, while fine-grained geometry, material properties, and other appearance cues can be safely discarded. This assumption is not fully justified, especially since many real-world manipulation tasks depend on friction, deformation, transparency, or subtle surface geometry that cannot be captured by a sparse primitive-based structure map. Furthermore, although the simulation results are extensive, the real-robot evaluation is comparatively limited, lacking ablation studies that would demonstrate whether the proposed structural representation actually improves robustness under realistic visual and physical variability. The method also does not provide evidence of generalization to unseen object categories or structures outside the template library used by StructGen, which raises concerns about how well the approach scales beyond curated environments. Finally, the full system relies on computationally heavy components such as Grounded-SAM and template retrieval, yet the paper reports only the added policy-side inference time; the end-to-end latency in closed-loop control is not addressed, leaving unclear whether the method can operate efficiently in real-time manipulation settings.

**Questions:**

1. Your approach assumes that a sparse structure map is sufficient to support manipulation, while fine geometry, surface appearance, and material cues are discarded. Can you provide empirical evidence or analysis showing that these omitted visual factors are not important for the tasks considered, especially in real-world settings?

2. The real-world experiments demonstrate successful executions but do not include ablations comparing structure-guided policies vs. baseline policies under real-world visual perturbations. Can you provide results isolating the contribution of the structure map in real-robot scenarios?

3. StructGen depends on template retrieval from a predefined object library. How does the system behave when encountering novel object categories or shapes that are not represented in the template set, and do you have experiments evaluating such cases?

4. The structure map is generated from simulation-trained primitives and template retrieval. When deploying to the real robot, how is sim-to-real transfer handled, especially given differences in object appearance, geometry, and environmental setup? Do you have qualitative or quantitative results demonstrating that the structure map remains accurate and stable when applied to real-world scenes, beyond the limited examples shown?

5. Grounded-SAM and template retrieval are computationally heavy modules. What is the end-to-end control-loop latency when StructGen is included, and does the structure map need to be recomputed at every control step? If not, how is temporal consistency maintained?

---

### Official Review · Reviewer_i84r · 2025-11-01

**Soundness:** 2
**Presentation:** 2
**Contribution:** 2
**Rating:** 4
**Confidence:** 4

**Summary:**

This work proposes to improve robustness in imitation learning by StructPolicy, a module that guides the robot to acquire structural knowledge by constructing a stable and task-relevant Structure Map, which is composed of task-relevant object structures and helps filtering out visually distracting noises. The authors introduce a general object structure representation based on atomic geometric primitives, and StructGen, a module that automatically constructs structure maps from visual observations. A StructTransformer model is designed to employ hierarchical attention over structure vertices to extract map features from the structure map and use for action prediction.

The authors evaluate StructPolicy across different imitation learning models, simulators, and manipulation tasks, and demonstrate that
StructPolicy consistently improves robustness against distribution shifts.

**Strengths:**

1. The problem formulation is valid and well-motivated. Despite recent advances in imitation learning methods for robotic manipulation, robustness to distribution shifts is still an open problem, which the authors attribute to the policy being overfit to task-irrelevant visual noises. The use of pre-trained visual models, i.e. Grounded-SAM, is a good approach in harnessing the visual robustness baked during their large-scale pretraining.

2. The experiment results span both simulation and real world setups. The reported empirical results demonstrate significant improvement in terms of robustness to visual noises and camera shifts. The paper is well-written and easy-to-follow.

**Weaknesses:**

1. It's unclear whether the combinations of Atomic Geometric Primitives (AGPs) is expressive or general enough to represent a broader variety of object geometries beyond those in the simulation tasks studied in the paper.

2. As I understand it, the StructMap representation requires knowledge of the overall task scene setup all the objects already known, so it's unclear whether/how the method would handle cases where an object was not initially observed in the scene but appear later in the task?

3. Although the authors have dedicated efforts into evaluating in well-established simulation benchmarks, the real world experiments are limited in scope. To properly study visual distribution shift, it would be more convincing if more tasks with real world images can be investigated.

**Questions:**

1. How would one use the proposed Atomic Geometric Primitives (AGPs) to represent more complex object geometries, e.g. a mug? And in the paper, does the robot arm's geometry get encoded by the StructGen representation?

2. Why are all the numbers bolded in Table 2? Why is it that none of the entries are bolded in Table 8? What's the takeaway from the overall results in these two tables?

3. In Figure 5 and policy rollout videos in supplementary materials, why are the camera views so zoomed into the table top? The drawer task has only a partial drawer visible, is this by design?

---

### Note · Authors · 2025-11-13

I have read and agree with the venue's withdrawal policy on behalf of myself and my co-authors.